# Kynurenic Acid Accelerates Healing of Corneal Epithelium In Vitro and In Vivo

**DOI:** 10.3390/ph14080753

**Published:** 2021-07-30

**Authors:** Anna Matysik-Woźniak, Waldemar A. Turski, Monika Turska, Roman Paduch, Mirosław Łańcut, Paweł Piwowarczyk, Mirosław Czuczwar, Robert Rejdak

**Affiliations:** 1Department of General and Pediatric Ophthalmology, Medical University of Lublin, 20-079 Lublin, Poland; rpaduch@poczta.umcs.lublin.pl (R.P.); robert.rejdak@umlub.pl (R.R.); 2Department of Experimental and Clinical Pharmacology, Medical University of Lublin, 20-090 Lublin, Poland; waldemarturski@umlub.pl; 3Department of Pharmacology, Faculty of Health Sciences, Medical University of Lublin, 20-093 Lublin, Poland; turskamk@gmail.com; 4Postgraduate School of Molecular Medicine, Medical University of Warsaw, 02-091 Warsaw, Poland; 5Department of Virology and Immunology, Institute of Microbiology and Biotechnology, Maria Curie-Skłodowska University, 20-033 Lublin, Poland; 6Center for Experimental Medicine, Medical University of Lublin, 20-090 Lublin, Poland; miroslaw.lancut@umlub.pl; 72nd Department of Anesthesiology and Intensive Care, Medical University of Lublin, 20-081 Lublin, Poland; piwowarczyk.pawel@gmail.com (P.P.); miroslaw.czuczwar@umlub.pl (M.C.)

**Keywords:** kynurenic acid, corneal epithelium, conjunctival epithelium, accelerated healing

## Abstract

Kynurenic acid (KYNA) is an endogenous compound with a multidirectional effect. It possesses antiapoptotic, anti-inflammatory, and antioxidative properties that may be beneficial in the treatment of corneal injuries. Moreover, KYNA has been used successfully to improve the healing outcome of skin wounds. The aim of the present study is to evaluate the effects of KYNA on corneal and conjunctival cells in vitro and the re-epithelization of corneal erosion in rabbits in vivo. Normal human corneal epithelial cell (10.014 pRSV-T) and conjunctival epithelial cell (HC0597) lines were used. Cellular metabolism, cell viability, transwell migration, and the secretion of IL-1β, IL-6, and IL-10 were determined. In rabbits, after corneal de-epithelization, eye drops containing 0.002% and 1% KYNA were applied five times a day until full recovery. KYNA decreased metabolism but did not affect the proliferation of the corneal epithelium. It decreased both the metabolism and proliferation of conjunctival epithelium. KYNA enhanced the migration of corneal but not conjunctival epithelial cells. KYNA reduced the secretion of IL-1β and IL-6 from the corneal epithelium, leaving IL-10 secretion unaffected. The release of all studied cytokines from the conjunctival epithelium exposed to KYNA was unchanged. KYNA at higher concentration accelerated the healing of the corneal epithelium. These favorable properties of KYNA suggest that KYNA containing topical pharmaceutical products can be used in the treatment of ocular surface diseases.

## 1. Introduction

One of the most important challenges in modern ophthalmic pharmacology is how to accelerate the healing of corneal epithelial defects and prevent the migration of conjunctival epithelium onto the cornea. In addition, the reduction of pain and prevention of scarring of the conjunctiva after surgery (especially anti-glaucoma surgery) or trauma will be welcomed. So far, many attempts have been considered in topical therapy of ocular surface including blood derivatives [1], saliva [2], conditioned media from human amniotic epithelial cells [3], the growth hormone [4], and erythropoietin [5], albeit without satisfactory outcomes.

In the search for an adequate candidate, we focused on endogenous compounds present in tears. In our previous studies, tryptophan (TRP) and its immediate metabolite kynurenine were investigated [6,7]. Kynurenic acid (KYNA) is another TRP metabolite formed along the kynurenine pathway. KYNA is an organic, water-soluble compound that was discovered in dog urine in 1853 by Justus von Liebig [8]. KYNA (C_10_H_7_NO_3;_ 4-hydroxyquinoline-2-carboxylic acid) has a molecular weight of 189.17 and solubility >28.4 ug/mL (https://pubchem.ncbi.nlm.nih.gov, accessed on 22 July 2021). Its chemical structure is shown in Scheme 1.

KYNA is widely distributed in nature. It is present in body fluids as blood, cerebrospinal fluid, saliva, and tissues including eye structures in mammals. We found that it occurs naturally in tears at a concentration of 0.02 μM [9]. It also occurs in plants, vegetables, and herbs as a component of everyday diet [10]. KYNA is considered as a safe and non-toxic substance that cannot cross the blood–brain barrier.

Endogenous KYNA is a product of normal TRP metabolism. More than 90% of available TRP is catabolized along the kynurenine pathway (KP) [11]. TRP is converted by the first and rate-limiting enzymes, tryptophan 2,3-dioxygenase constitutively present in liver and cytokine-activated indoleamine-2,3-dioxygenase (IDO) broadly distributed in body tissues to *N*-formyl kynurenine, which is metabolized to kynurenine (KYN) by kynurenine formamidase. KYN is mainly converted to 3-hydroxykynurenine and further downstream metabolites. KYNA synthesis is catalyzed by kynurenine aminotransferases (KATs) I-IV. Previously, we demonstrated the presence of KAT I, II, and III in the healthy human limbal conjunctiva and the cornea. Our findings support the hypothesis that TRP can be metabolized to KYNA in corneal epithelium, endothelium, and stroma [12]. KYNA displays many biological activities. KYNA exhibits anti-inflammatory, antioxidative, analgesic, and antimicrobial properties [13,14,15,16]. It is the only known naturally occurring antagonist of endogenous glutamate receptors in the central nervous system. KYNA in low micromolar concentrations antagonizes the glycine site of the *N*-methyl-D-aspartate (NMDA) receptor complex and acts as a potent neuroprotectant. As a result of this activity, it may influence important neurophysiologic and neuropathologic processes including Alzheimer’s disease, Huntington’s and Parkinson’s diseases, multiple sclerosis, epilepsy, brain ischemia, depression, and schizophrenia. Glutamate in high concentration is known for its neurotoxicity in the brain and retina [17,18]. In a similar manner, it may play a role in neurotrophic diseases of cornea and the remaining ocular surface structures. In the cornea, glutamate interacts also with its receptors to cause nociception. This suggests that the blockade of these receptors may help in controlling inflammatory or maladaptive pain from the cornea [19].

Furthermore, it has been shown that KYNA is an endogenous antagonist of the alpha7 nicotinic acetylcholine receptor, G protein-coupled receptor 35 (GPR35), and aryl hydrocarbon receptor (AhR) ligand [20,21]. The presence of GPR35 in human corneas has been revealed recently; however, its role in the eye requires further investigation [22]. Accumulated data suggest that GPR35 may play an important role in response to hypoxic stress and be a potential target for the treatment of inflammatory, allergic, and neurological disorders [23]. The potential role of AhR in ocular and non-ocular neurodegenerative diseases has been recently reviewed [24]. The beneficial effects of KYNA in a number of pathological conditions have been reported, but no studies have been conducted on its use in the treatment of eye diseases so far.

Since the effect of KYNA on structures of the ocular surface has never been explored in the present study, the influence of KYNA on corneal and conjunctival cells in vitro and the re-epithelization of corneal erosion in rabbits in vivo was investigated.

## 2. Results

### 2.1. Effect of KYNA on Metabolic Activity of Corneal and Conjunctival Cells In Vitro

The incubation of both corneal epithelial cells and conjunctival epithelial cells in medium containing KYNA at concentrations of 1–100 μM for 24 h and 48 h resulted in a concentration-dependent reduction of metabolic activity of these cells measured with a 3-[4,5-dimethylthiazol-2-yl]-2,5 diphenyl tetrazolium bromide (MTT) assay (Figure 1a,b).

### 2.2. Effect of KYNA on Viability of Corneal and Conjunctival Cells In Vitro

The presence of KYNA (1–100 μM) in the medium did not affect the viability of corneal epithelial cells measured after incubation lasting for 24 h and 48 h using a neutral red (NR) assay (Figure 2a). KYNA (1–100 μM) present in the incubation medium transiently reduced the viability of conjunctival epithelial cells. The viability decline was significant after 24 h but not after 48 h of incubation. The effect was not concentration-dependent.

### 2.3. Effect of KYNA on Cellular Cytoskeleton F-Actin Organization of Corneal and Conjunctival Cells

The following parameters were visually assessed: cell morphology, based on the shape, structure, form, and size of cells and the borderline of the cytoplasmic membrane; condition of the cell membrane, based on its form and linearity; intercellular interactions and adhesion of cells to the culture surface.

KYNA (10 μM, 100 μM) did not affect cell morphology in the corneal epithelial cells (Figure 3a) and conjunctival epithelial cells (Figure 3b). In the corneal and conjunctival epithelial cells, the F-active filaments of the cytoskeleton were located mainly near the cytoplasmic membrane. In conjunctival epithelial cells, high concentration of the fibers was also observed in the vicinity of the nuclear membrane.

KYNA (10 μM, 100 μM) did not affect intercellular adhesions in the corneal epithelial cells (Figure 3a). In conjunctival epithelial cells incubated in the presence of KYNA at both concentrations, a looser arrangement of cells may indicate a weakening of intercellular connections and, indirectly, a change in the structure of F-actin fibers after the exposure of cells to KYNA (Figure 3b).

### 2.4. Effect of KYNA on Transwell Migration Capacity of Corneal and Conjunctival Cells In Vitro

In the presence of KYNA at a concentration of 75 μM for 48 h, the migration capacity of corneal epithelial cells was elevated to 208% (*p* < 0.05), whereas that of conjunctival epithelial cells was not significantly enhanced up to 120% (*p* > 0.05) as compared to control (set as 100%) (Table 1).

### 2.5. Effect of KYNA on Cytokine Release by Corneal and Conjunctival Cells In Vitro

Obtained results demonstrated that KYNA stimulates corneal and conjunctival epithelial cells to secret cytokines, thus pointing to its mediatory role in immune response.

The incubation of corneal epithelial cells in a medium containing KYNA at concentrations of 1 μM and 10 μM for 24 h and 48 h did not significantly affect the release of IL-1β except for the decreased release of this cytokine after 24 h of cell incubation with KYNA 10 μM (Figure 4a).

The incubation of conjunctival epithelial cells in a medium containing KYNA at a concentration of 1 μM significantly reduced the release of IL-1β after 24 h and 48 h incubation. KYNA at a higher concentration (10 μM) was ineffective in this regard (Figure 4b).

The release of IL-6 was reduced and enhanced by KYNA 10 μM from corneal epithelial cells incubated for 24 h and 48 h, respectively (Figure 4c). The release of IL-6 was enhanced by KYNA 1 μM and 10 μM from conjunctival epithelial cells incubated for 24 h and 48 h, respectively (Figure 4d).

The release of IL-10 from corneal epithelial cells was unaffected by the presence of KYNA in the medium at both concentrations studied: 1 μM and 10 μM, which were measured after 24 h and 48 h of incubation (Figure 4e). The release of IL-10 from conjunctival epithelial cells was unaffected by the presence of KYNA in the medium at both concentrations studied: 1 μM and 10 μM, which were measured after 24 h and 48 h of incubation except for increased release of this cytokine after 48 h of cell incubation with KYNA 10 μM (Figure 4f).

### 2.6. Effect of KYNA on Corneal Epithelialization In Vivo

Eye drops containing KYNA were administered five times daily for 3 days. The administration of eye drops containing KYNA at a concentration of 0.002%, which corresponds to the highest concentration used in in vitro experiments (100 μM), did not affect the duration of healing of experimentally induced erosion of the cornea in rabbits (data not shown). The administration of eye drops containing KYNA at a concentration of 1% significantly shortened the duration of healing of experimentally induced erosion of the cornea in rabbits (Figure 5a; Table 2). The dynamics of the re-epithelization process was unaffected by KYNA up to 24 h after insult and significantly accelerated thereafter as compared to control (Figure 5b). Careful examinations performed throughout the entire study by two experts, an authorized veterinarian and an experienced human ophthalmologist, did not reveal any adverse effects of tested drops. No signs of discomfort of animals were found. The condition of cornea and conjunctiva was evaluated using a biomicroscope (a slit lamp examination without and with cobalt blue filter after fluorescein administration). The appearance of these structures was documented photographically at the time of wound area measurement, i.e., every 12 h for 3 days for the whole period of applications of eye drops.

## 3. Discussion

In this study, for the first time, we showed that KYNA accelerates the re-epithelization of experimentally induced corneal erosion in rabbits in vivo. The administration of KYNA containing eye drops accelerated the process of re-epithelization 36 h after surgery, leading to an earlier full recovery of the corneal surface. This observation is consistent with the results of in vitro experiments in which KYNA promoted the migration of human corneal epithelium. This conclusion is consistent with the finding of Poormasjedi-Meibod et al. (2014) that KYNA accelerated keratinocyte migration, as measured in scratch assay in vitro [25]. The molecular mechanisms responsible for the healing properties of KYNA is unclear. KYNA is a NMDA receptor antagonist. However, according to Oswald et al. (2012), activation rather than inhibition of the NR1 subunit of the NMDA receptor in concert with purinergic mechanisms stimulates the healing processes in a co-culture of human corneal limbal epithelial cells, and primary trigeminal neurons and NMDA antagonists expressed opposite effects [26]. Interestingly, KYNA is also an agonist of the AhR and GPR35 receptors [27]. Although the presence of AhR in cornea was established, it is unclear whether this receptor participates in repair processes [28]. Ikuta et al. 2008 did not find any significant difference in skin wound closure between wild-type and AhRR-/- mice. On the contrary, they found that the inactivation of AhR accelerates wound healing in the early inflammatory phase of this process [29]. This result has to be treated with caution, since a substantial lack of homology between AhR signaling in mice and in humans was reported, which is due to very low sequences homology in the C-terminal transactivation domain of AhR [30,31]. Ultimately, the healing effect of KYNA could be explained by the activation of GPR35 in the corneal epithelium. Recently, the presence and distribution of GPR35 receptors in the human cornea was demonstrated [22]. Furthermore, Tsukahara et al. (2017) reported that GPR35 agonists promoted wound repair of mouse colon epithelium [32]. The indirect effect of KYNA should be also considered. It has been shown that the process of corneal healing is complex, and many neuropeptides/neurotransmitters and receptors are involved in cell proliferation and migration [33]. Thus, the anti-inflammatory and immunosuppressive functions of KYNA may play a role in regulating healing processes as well [34].

Although KYNA reduced cell metabolism, it did not affect the viability of corneal epithelium in vitro. Additionally, it did not influence the cytoskeleton and cell–cell junctions in the corneal epithelial cells. Accordingly, the lack of toxic effects of KYNA applied in a wide range of concentrations was confirmed in our previous study, in which an HCE three-dimensional model consisting of immortalized human corneal epithelial cells was utilized, pointing to the safety of KYNA application on the eye surface [6].

Another aspect of healing corneal epithelial wounds in vivo is the unwanted migration of conjunctival epithelial cells across the denuded limbus to cover the corneal: the process termed conjunctivalization [35]. It was reported that the area of cornea covered by the conjunctival epithelium appeared dysfunctional, thin, and irregular; it attracted new vessels and was prone to recurrent erosions. Moreover, conjunctivalization of the visual axis affected vision; therefore, the mechanical removal of the conjunctival epithelium from the cornea is recommended [35]. Intriguingly, we found that KYNA impairs both the metabolism and viability of the conjunctival epithelium and influences intercellular adhesions among cells in vitro. In contrast to its effect exerted on the corneal epithelium, KYNA does not enhance the migratory capacity of the cells in conjunctival epithelium. Such properties (see Table 3) imply that KYNA could reduce the severity or prevent the conjunctivalization of cornea. This assumption needs confirmation in an appropriate animal model and/or clinical trial.

In recent years, numerous in vivo and in vitro studies have been directed toward the immunomodulatory functions of KYNA. Therefore, we studied the release of cytokines IL-1β, IL-6, and IL-10 from corneal and conjunctival cells in culture exposed to KYNA. We found that KYNA-induced changes in cytokine production varied markedly by the cell type, drug concentration, and incubation time. We found that KYNA enhanced the release of IL-1β from corneal and conjunctival cells, and this result agreed with data, presenting an increased release of cytokines from the corneal epithelium during injuries [36]. KYNA enhanced the release of IL-6 from corneal and conjunctival cells. This pro-inflammatory cytokine is thought to play an important role in corneal epithelial wound closure in vivo and to stimulate the migration of human corneal epithelial cells in vitro [37,38,39]. We also found that in a conjunctival epithelium culture, KYNA enhanced the release of IL-10; the cytokine is known to promote anti-inflammatory and anti-angiogenic processes in the cornea and conjunctiva [40] as well as inducing corneal transplantation immune tolerance [41]. Our results suggest that the KYNA-induced release of cytokines may affect the wound healing of the cornea, since it was shown by Arranz-Valsero et al. (2014) that the presence of IL-6 or IL-10 increased the wound-healing rate in an in vitro corneal wound-healing model [38].

Furthermore, KYNA possesses other favorable properties of potential beneficial applications in ophthalmology. Csati et al. (2015) [42] have shown that KYNA reduced experimentally induced inflammation in the mandibular part of the trigeminal ganglion. Since the administration of KYNA resulted in the inhibition of the signaling system of the trigeminal ganglion, it could be speculated that KYNA may reduce corneal pain without nerve damage. This assumption is strengthened by the evidence on the antimigraine and pain-relieving activity of KYNA and its analogues [43,44] The antinociceptive properties of KYNA topically administered to the conjunctival sac have yet to be established.

The antiscarring properties of topically applied KYNA in a skin wound-healing model in rabbits and rats were reported [25,45]. Importantly, KYNA enhanced the expression of matrix metalloproteinases MMP1 and MMP3 and suppressed the production of type I collagen and fibronectin by fibroblasts [25]. Based on these results, it could be assumed that topically administered KYNA may reduce conjunctival and scleral scarring after glaucoma surgery, which is a major limiting factor to the success of this procedure [46]. Moreover, the use of KYNA to inhibit the overgrowth of stromal fibroblasts and aberrant matrix remodeling in pterygium also could be postulated. It was revealed that KYNA protected endothelial cells in culture against homocysteine-induced cytotoxicity [47]. Increased homocysteine levels in tears and plasma were found in patients with glaucoma [48]. A link between high homocysteine and other eye diseases such as retinopathy, maculopathy, cataract, optic atrophy, and retinal vessel atherosclerosis is suggested [49].

KYNA is an endogenous substance and diet constituent but it is not approved as a drug. It is important to note in this regard that the results of a phase 1 clinical trial of topically administered KYNA were published recently [50]. KYNA was applied to the skin of volunteers at concentrations ranging from 0.15% to 0.5%. The skin reaction was estimated 24 h later. The procedure was repeated after 14 days to establish a potential delayed allergic reaction. No adverse reactions were recorded. Moreover, no sign of absorption of KYNA from the skin was recorded in blood and urine [50].

Interestingly, KYNA was identified as UV filter [51]. Thus, its presence in a lens might have an additional beneficial effect; i.e., protection of the lens and retina from photodamage. On the other hand, it was found that KYNA is the most photochemically active dye of the human eye lens reported to date [52]. In the lens, under physiological conditions, KYNA is formed from kynurenines undergoing thermal and photochemical reactions [53]. It was suggested that due to the high yield of the triplet state of its active form, it may contribute to the photodamage of lens proteins [52]. Thus, its involvement in cataract development cannot be excluded. In fact, elevated content of KYNA was detected in cataractous lenses [54]. The question remains open of whether KYNA applied topically may enhance the content of KYNA in its active states in lenses? The penetration of KYNA through the cornea after topical administration is unknown, as well as its concentration in the aqueous humor of the anterior chamber in healthy eyes. However, the presence of endogenous KYNA was detected in the aqueous humor of the anterior chamber of cataractous eyes [55].

In conclusion, KYNA showed no toxicity to human conjunctival and corneal epithelium in a wide range of concentrations. Its action profile—accelerated healing of corneal epithelium with simultaneous lack of stimulation of conjunctival epithelial cells—seems to be beneficial from the point of view of treatment of many ocular surface disorders. KYNA is a compound with a multidirectional effect proposed for the treatment of neurological diseases. However, it was hampered by the fact that KYNA does not cross the blood–brain barrier. It is necessary to undertake further studies on the effect of KYNA on other types of cells present in structures forming the ocular surface. Summing up, all the above-described features make KYNA a drug candidate with wide therapeutic potential in the therapy of ophthalmic diseases.

## 4. Materials and Methods

### 4.1. Materials

KYNA was purchased from Sigma-Aldrich (St. Louis, MO, USA). The purity of the KYNA was ≥98%. Substance was used without further purification.

### 4.2. In Vitro Studies

#### 4.2.1. Cell Cultures

A human normal corneal epithelial cell line 10.014 pRSV-T (ATCC No. CRL-11515) and human normal epithelial conjunctival cell line (ATCC No. CRL-12658) were used. Cells were cultured as monolayers in 25 cm^2^ culture flasks (Nunc. Roskilde, Denmark) coated with PureCol^TM^ ultrapure collagen (INAMED Biomaterials, Fremont, CA, USA) at 3.1 mg/mL concentration. One mL of collagen was poured into the bottles or plate wells, spread over the entire surface, and after 3 min, the excess solution was removed. The remaining collagen was dried in a thermostat at 37 °C. The cell line was maintained in defined K-SFM (keratinocyte-serum free medium) (Gibco^TM^, Paisley, UK) supplemented with the 75 μg/mL endothelial cell growth factor (ECGF) (Sigma, St. Louis, MO, USA), 0.05 mg/mL bovine pituitary extract (BPE) (Gibco), 500 ng/mL hydrocortisone (Sigma), and 0.0005 mg/mL bovine insulin (Gibco), and antibiotics (100 U/mL penicillin, 100 μg/mL streptomycin) (Sigma, St Louis, MO) at 37 °C in a humidified atmosphere with 5% CO_2_.

#### 4.2.2. Experimental Schedule

Human corneal and conjunctival epithelial cell suspension (1 × 10^5^ cells/mL) in a culture medium was added to the appropriate culture dishes (96-well or 24-well) and incubated for 24 h at 37 °C in a humidified 5% CO_2_/95% air incubator. Thereafter, the medium was replaced with a new one containing appropriate concentrations of KYNA (0–100 μM). A blank control consisted only of culture medium. Incubation of cells (1 × 10^5^ cells/mL) with KYNA was continued for 24 h and 48 h.

#### 4.2.3. Cell Metabolism and Viability—MTT Assay

The 3-[4,5-dimethylthiazol-2-yl]-2,5 diphenyl tetrazolium bromide (MTT) assay is based on the conversion of a yellow tetrazolium salt by viable cells to purple crystals of formazan. The reaction is catalyzed by mitochondrial succinate dehydrogenase. Cells grown in 96-well multiplates in 100 μL of culture medium for 24 h or 48 h of incubation with KYNA were further incubated for 3 h with MTT solution (5 mg/mL, 25 μL/well) (Sigma-Aldrich). The crystals of formazan were solubilized overnight in a 10% sodium dodecyl sulfate (SDS) in 0.01M HCl. The product was quantified spectrophotometrically by absorbance measurements at 570 nm using an E-max Microplate Reader (Molecular Devices Corporation; Menlo Park, CA, USA).

#### 4.2.4. Cell Viability-Neutral Red (NR) Uptake Assay

The NR cytotoxicity assay is based on the uptake and lysosomal accumulation of the neutral red dye. Dead or damaged cells do not take up the dye. Cells were grown in 96-well multiplates in 100 μL of culture medium (K-SFM) with KYNA. Subsequently, the medium was discarded, and 0.4% NR (Sigma) solution was added to each well. The plate was incubated for a further 3 h at 37 °C in a humidified 5% CO_2_/95% air incubator. After incubation, the dye-containing medium was removed, cells were fixed with 1% CaCl_2_ in 4% paraformaldehyde, and the incorporated dye was solubilized using 1% acetic acetate in a 50% ethanol solution (100 μL). The plates were gently shaken for 20 min at room temperature, and the extracted dye absorbance was measured spectrophotometrically at 540 nm.

#### 4.2.5. Cellular Cytoskeleton F-Actin Organization Analysis

Cells were incubated in 4-well Lab-Tek chamber slides (Nunc) filled with 1 mL of culture medium supplemented with KYNA at a 10 or 100 μM concentration. After incubation, the cells were rinsed with K-SFM medium and exposed to paraformaldehyde (10%, *v*/*v*) solution for 20 min., rinsed three times in PBS, exposed to Triton X-100 (0.2%, *v*/*v*) (Sigma) solution for 5 min, and rinsed three times with PBS. Then, 0.5 mL PBS containing tetramethyl-rhodamine-isothiocyanatephalloidin (TRITC-phalloidin, 1 μg/mL) (Sigma) was added to each well and incubated in the dark at 37 °C/5% CO_2_ for 30 min. Cell analysis was conducted under a fluorescent microscope (Olympus, BX51, Tokyo, Japan). A qualitative assessment of fluorescent images was performed using the AnalySIS imaging software system.

#### 4.2.6. Transwell Migration Test

The analysis was performed in 24-well plates with inserts (0.4 μm pore membrane) (Nunc). Corneal and conjunctival epithelial cells (1 × 10^5^ cells/mL) were seeded into the upper chambers. Culture medium was placed into the upper chamber (150 μL), while the medium in the lower chamber (600 μL) contained KYNA at 75 μM concentration. The experiment was conducted for 48 h at 37 °C in a humidified atmosphere with 5% CO_2_. Cytoplasmic projections through the membrane’s pores were fixed with 4% paraformaldehyde for 10 min and stained with a Giemsa stain (Sigma) for 20 min. Cells adhered to the upper surface of the membrane were removed using a cotton swab. In order to quantify the number of migrating cells, the stained parts of cells (pseudopodia) were counted under a microscope (Olympus BX51, Tokyo, Japan) at 40× magnification.

#### 4.2.7. ELISA Assay

The level of human IL-1β, IL-6, and IL-10 was measured by an *immunoenzymatic* method (ELISA) using a commercially available kit (BD OptEIATM, San Jose, CA, USA) according to the manufacturer’s instruction. The optical density set at 450 nm was determined using a microplate reader (Molecular Devices Corp., Emax, Menlo Park, CA, USA). The concentration of the cytokines was calculated based on the standard curve. The detection limit was 0.8 pg/mL for IL-1β, 2.2 pg/mL for IL-6, and 2 pg/mL for IL-10.

### 4.3. In Vivo Studies

#### 4.3.1. Animals

Sixteen healthy New Zealand white rabbits of both sexes weighing between 3.0 and 3.3 kg were used. The animals were kept in single cages upon arrival in the facilities under standard conditions of temperature (21 °C) and 12:12 h light/dark cycle, with food and tap water provided ad libitum. After a week of adaptation in the facility, the animals were admitted to the experimental session. To exclude any disease that could interfere with the re-epithelization process, the eyes of all animals were previously examined with a slit lamp. The rabbits were kept in restraining boxes only during the experiments.

#### 4.3.2. Anesthesia and Surgical Procedure

The rabbits were anesthetized with an intramuscular injection of 1.2–1.8 mL of a ketamine/xylocaine mixture (7:1, *v*:*v*). All of them underwent a bilateral corneal de-epithelialization procedure according to Moshirfar et al. [56]. The surgery was performed by an experienced surgeon as follows. After a wire lid speculum was installed, approximately 0.5 mL of 20% ethanol was placed on the central cornea within the barrel of a 6 mm marking trephine. After 60 s, the ethanol was removed with a cellulose sponge, and the surface of the eye was irrigated with 0.9% saline solution for 60 s. The ethanol-exposed epithelium was removed with a saline-moistened sponge or a spatula when necessary. Then, the same procedure was repeated on the contralateral eye.

#### 4.3.3. Treatment

Postoperatively, each rabbit received a solution of KYNA (0.002% or 1%) and 0.9% saline solution in the form of eye drops in the right and left eye, respectively. The schedule for the designated drop medication was 5 times a day, up to 72 h, or until complete epithelial healing occurred. The first dose was applied immediately after de-epithelialization of the cornea and documentation of the lesion size (time 0). To obtain 0.002% solution, KYNA was dissolved in 0.9% saline at room temperature. To obtain a 1% solution of KYNA, it was necessary to dissolve it in 1N NaOH until the substance was completely dissolved. Then, saline was added, and the solution was adjusted with HCl to pH approximately 7.5.

#### 4.3.4. Evaluation of Corneal Epithelialization

The lesions of both eyes were examined immediately (time 0) and every 12 h postoperatively. This involved the application of fluorescein (BIO-GLO, Hub Pharmaceuticals LCC, Rancho Cucamonga CA, USA) into the eye, without topical anesthesia, followed by a slit lamp examination under cobalt blue light. Digital images were taken with a Canon EOS 450D 12.2 megapixel camera (Canon Inc., Tokyo, Japan), coupled to a YZ5X1 slit lamp (66 Vision-Tech Co., Ltd., Suzhou, China).

#### 4.3.5. Image Analysis System

The digital images were evaluated using the EPCO 2000 (Prof. Dr M. Tetz, Augentagesklinik Spreebogen, Germany) computer analysis system, which offers greater precision than slit lamp measurements. The EPCO 2000 computerized system (developed by Tetz et al. [57]) was originally created for the evaluation of posterior capsule opacification. However, it has already been used for the analysis of corneal epithelial defect surface areas in rabbits as well [56]. The area of the corneal epithelial defect was outlined with a hand-held pointing device and measured in pixels. The re-epithelization rate was assessed as a ratio of the actual epithelial defect area to its initial size at time zero (percentage of the control value).

### 4.4. Statistical Analysis

Results of in vitro experiments (n = 3) and in vivo experiments (n = 8) are presented as a mean ± standard deviation (SD) or standard error of the mean (SEM). Statistical analysis was performed using one-way analysis of variance ANOVA followed by Dunnett’s multiple comparison post hoc test. Differences of *p* < 0.05 were considered significant.

## Data Availability

Cell lines are available in ATCC. The rabbits came from a licensed farm in Chorzelów near Kraków, which is in the register of the Ministry of Science and Higher Education of Poland. The datasets generated during and/or analyzed during the current study are available from the author (Anna Matysik-Woźniak) on reasonable request.

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
