# Peer review of "Kynurenic Acid Accelerates Healing of Corneal Epithelium In Vitro and In Vivo"

_pharmaceuticals, 2021, doi:10.3390/ph14080753_

Round 1

Reviewer 1 Report

In this report, Matysik-Wozniak et al. evaluate the effects of kynurenic acid (KYNA) on some basic cellular phenotypes in conjunctival and corneal epithelial cells, as well as determine the effects of KYNA on corneal healing in vivo.

Authors demonstrate interesting findings on the cellular effects of KYNA, and thus this paper is of interest. However, the authors need to better explain the meaning of their results for the reader, and better guide the reader through their interpretations of their data. There are also several inconsistencies within the paper as well (explained in more detail below). Lastly, the positive findings of this manuscript are largely surface level. More detailed analysis of the in vivo experiments in Figure 5 (and quantified in Table 2) would lead to greatly heightened impact of the manuscript. 

Specific points:

Authors should proofread for grammatical errors, unclear statements, etc. throughout the manuscript.

Authors should make the title one statement. Consider “Kynurenic acid accelerates healing of corneal epithelium in vitro and in vivo”

Abstract should include a little bit more background information; it’s not obvious from reading it why the authors would think to test KYNA on corneal and conjunctival cells. The connection should be clear to the reader from looking at the abstract.

In the manuscript figures, it would be better for the reader if instead of labelling “pRSV-T” and “HC0597”, the authors labelled them corneal epithelial cells and conjunctival epithelial cells.

Authors should provide brief interpretation of their results in the results section when relevant. For example, the authors note that KYNA reduces MTT metabolism in pRSV-T cells but does not affect their viability using a neutral red assay over the same time frame. This seems to suggest that KYNA is reducing cellular proliferation.

2.3 the authors state that “KYNA… did not affect F-actin filaments organization and intracellular adhesions in the corneal epithelial cells.” Authors need to be much more specific here. What metrics and parameters did they use to come to this conclusion? It is difficult to replicate their conclusion using the images and vague description provided by the authors. In the methods the authors mention that “a quantitative assessment of fluorescent images was preformed using the AnalySIS imaging software system” but it is still not obvious what this means.

2.4 Why did the authors choose 75uM KYNA?

2.4 The authors’ text describing their findings here is confusing. The pseudopodia are being counted in order to quantify the number of migrating cells in the transwell assay. Therefore, the authors should just describe this as number of cell counts instead of the rate of pseudopodia formation. That information (pseudopodia) is best left for the methods section.

2.5 “Incubation of …did not significantly affect the release of IL-1B except for the decreased release of this cytokine after 24 h of cell incubation with KYNA 10uM. (Fig. 4a).” The authors do not show an * to denote statistical significance of this comparison (24h 10uM KYNA)

In 2.5, the authors list the changes upon KYNA stimulation of their epithelial cells, but it’s hard to see what conclusions can be drawn from the data. The authors should guide the reader more towards their interpretation of the data.

2.6 “The dynamics of re-epithelialization process was unaffected by KYNA up to 24 h after insult and significantly accelerated thereafter” But the authors only show an * denoting statistical significance in the table at t=36h (not at 48h).

If authors are oping not to show their negative data on healing for .002% KYNA eye drops, they should state (data not shown) for this experiment.

More information about the rabbit experiments should be included in the results section. Are the eye drops being administered just once? Are they being administered at the time of surgery? Etc. there should  be more information here to aid in basic interpretation of the authors’ experiments without having to read the full methods.

Since authors detected some pro-migratory effects of KYNA, it is worth the authors mentioning that this is consistent with Poormasjedi-Meibod et al. PLOS One e91955 (2014) in which authors demonstrated that KYNA accelerated keratinocyte migration via scratch assay (Figure 6).

The authors discussion regarding the effects of AhR on healing as assessed by studies in the knockout mouse is good and useful. However, the authors should be aware of the substantial lack of homology between AhR signaling in mice and in humans, due to very low sequences homology in the C-terminal transactivation domain of AhR. See Flaveny et al. “Differential gene regulation by the human and mouse aryl hydrocarbon receptor” (2010, Toxicological Sciences) and Flaveny et al “The mouse and human Ah receptor differ in recognition nof LXXLL motifs” (Archives of Biochemistry and Biophysics, 2008). This may potentially explain some contradictory findings among species.  

In the discussion authors state that “…..we found that KYNA….restructures (rearranges) the cytoskeleton of conjunctival cells by reducing intracellular adhesions among cells in vitro. But in the results the authors stated “KYNA did not affect F-actin filaments organization and intracellular adhesions in the corneal epithelial cells and conjunctival epithelial cells.” These statements are inconsistent.

Throughout the manuscript, mentions of pseudopodia formation are probably better described as migratory activity (that is the more relevant phenotype; pseudopodia formation is being used as a surrogate to determine migration).

Authors cite [43] regarding the anti-scarring properties of KYNA. It’s also worth citing and mentioning one of the follow-up articles by the same group “development of a nanofibrous wound dressing with an antifibrogenic properties in vitro and in vivo model” in JBMR (2016).

Methods:

“…coated with PureCol…at 3.1mg/mL concentration.” That is the concentration of the PureCol, I assume, but how was the flask coated?

“Postoperatively, each rabbit received a solution of KYNA (.1% or 1%).” Authors previously mentioned using 0.002% KYNA in the results section.

Reviewer 2 Report

Below, there is the review of the manuscript (pharmaceuticals-1306805-peer-review-v1) entitled "Kynurenic acid accelerates the healing of corneal epithelium. In vitro and in vivo studies."

The work presented here is a continuation of the Authors' research on the effects of kynurenic acid on the corneal and conjunctival epithelium and the potential use for treating their damage, which was presented previously in Pharmacological Reports 69 (2017) 722–729. Presently, the Authors conducted some new tests in vitro and additionally in vivo.

  1. The introduction should be more informative. Precise, please, information about the structure of the tested compound (kynurenic acid) should be included. Activities of chemical molecules are strictly connected with their structures. It is suggested to also include the figure with its structure and schemes of presented transformations of chemical individuals. Chemical names of a nitro substituted compounds should be presented in form with "N" in italic, i.e. for example "N-methyl-D-aspartate" should be instead of "N-methyl-D-aspartate". The scientific facts should be supported by providing appropriate references, for example after the statement "KYNA is an organic, water-soluble compound which was discovered in dog urine in 1853 by Justus von Liebig."
  2. In the Results section. The MTT cytotoxicity assay shows an obvious toxic effect with an increasing concentration on both cell types (human corneal and human conjunctival cells). Statistically significant toxicity can already be seen at the lowest tested concentration i.e. 1 µM, which is approximately equal to 0.002% concentration (assuming a density of 1g/ml for such diluted aqueous solutions). From figure 1 it can be seen that at 100 µM (i.e. approximately about 0.2% solution) toxicity of investigated acid is approximately at 40%-50% level. In an in vivo study, the Authors examined the effects of the compound at 0.002% and 1% concentrations. A 1% water solution of kynurenic acid is approximately equal to its 500 µM solution. However, the Authors did not test the cytotoxicity of this acid at this concentration in vitro tests. Besides, treatment results in rabbits are not spectacular at this concentration (table 2). Generally, since the concentration of 1% (i.e. 500 µM) turned out to be the lowest concentration at which KYNA showed a statistically significant effect in vivo, the in vitro study of the effect of KYNA on cells should be additionally performed with this concentration to determine the effect of such concentration on tested cells. Furthermore, how is it possible that the compound shows statistically significant cytotoxicity in the NR-cytotoxicity assay on HC0597 cells after 24 h, but no longer after 48h? Can you explain it? This should be explained in detail.
  3. Why were 0.002% and 1% chosen for in vivo testing? If a concentration as small as 0.002% was found to be completely ineffective, why was 1%, a concentration 500 times greater than 0.002%, chosen? Have studies been done with intermediate concentrations, between 0.002 and 1%?
  4. In the description of 4.3.2. "Treatment", it is stated that KYNA solution of 0.1% or 1% was applied to rabbits, while in the discussion of results (section 2.6), the Authors report the concentrations of 0.002% and 1% (table 2). Can you please explain this discrepancy? Furthermore, there is no information on how the KYNA solutions were prepared. Besides, what is the pH of these solutions? The optimal pH should be close to the tear fluid, i.e. 7.0=7.4, in order to eliminate the additional irritating effect. Saline solution (imprecise term) - in what concentration? 0.9% (isotonic) saline solution is called "normal saline solution". Please also clarify whether the respective KYNA solutions were made by dissolving the appropriate amount of acid in saline and administered to the rabbits as such, or whether the acid solutions were administered independently of the saline solution.
  5. In section 2.4. To test the effect of kynurenic acid on the transwell migration capacity of corneal and conjunctival cells, the Authors used only one concentration of 75 µM. What is the justification for using just such concentration in this assay and not others?
  6. In Section 2.5, the Authors reported that kynurenic acid at a concentration of 10 µM decreases the release of the cytokine Il-1 after 24 h, but figure 4a shows that this effect is statistically insignificant. Furthermore, figure 4a indicates that there is even a statistically significant increase in Il-1 production under KYNA acid after 48 h. It is suggested the Authors should present all data in the form of numerical values, e.g. given in supplementary material in order to reproduce the statistical calculations by the potential readers of the article.
  7. In section 4 Materials and methods 4.1 Materials Specify the purity of purchased kynurenic acid and whether it was further purified before use in biological tests.
  8. The Authors should consider recent studies on the potentially harmful effects of kynurenic acid on the lens of the eye (Yuliya S. Zhuravleva, Peter S. Sherin, Influence of pH on radical reactions between kynurenic acid and aminoacids tryptophan and tyrosine. Part I. Amino acids in free state, Free Radical Biology and Medicine, 172 (2021) 331–339.).
  9. The discussion section should be redrafted in a clearer and more readable way. The authors refer to the research of other researchers in an unclear manner, mixing their own results with the results of others, which makes it difficult to easily and quickly assess the authors' own contribution (for example "We also found…….[ref. 38]…...[ref. 39]. Our results suggest that KYNA…….[ref. 36]"). First, the state of the current research on KYNA should be provided, and then what is new about the research of the authors of the presented work in relation to the previous results.
  10. At final, the Authors should improve the presentation, the current one seems to be too chaotic and therefore illegible and unclear.

Round 2

Reviewer 1 Report

The manuscript has been clarified and improved to my satisfaction. 

Reviewer 2 Report

Below, there is the review of the revised manuscript (pharmaceuticals-1306805-peer-review-v2) entitled "Kynurenic acid accelerates healing of corneal epithelium in vitro and in vivo studies."

  1. In the explanation, the Authors state that "(b) it is known from earlier publications that toxicities of topical drugs used to treat glaucoma in humans are at least 1000-times higher in vivo than in vitro:"

In fact, the opposite is true. At the same concentration, the compounds will be more toxic to cells in vitro than in vivo, especially when administered topically or externally considering at least the effect associated with incomplete absorption and other losses of the active ingredient.

  1. Besides, despite attempts, the Authors did not clearly explain why the NR-cytotoxicity assay on HC0597 cells shows a cytotoxic effect after 24 h and no such effect after 48 h.

Overall, the Authors have made significant improvements to the manuscript and have clarified most of the uncertainties in detail. To sum up I believe that the manuscript deserves to be accepted for publication in Pharmaceuticals journal.